# Prevalence, Risk Factors, and Complications of Oropharyngeal Dysphagia in Older Patients with Dementia

**DOI:** 10.3390/nu12030863

**Published:** 2020-03-24

**Authors:** Mᵃ Carmen Espinosa-Val, Alberto Martín-Martínez, Mercè Graupera, Olivia Arias, Amparo Elvira, Mateu Cabré, Elisabet Palomera, Mireia Bolívar-Prados, Pere Clavé, Omar Ortega

**Affiliations:** 1Geriatric department, Hospital de Sant Jaume i Santa Magdalena, Consorci Sanitari del Maresme, 08304 Mataro, Spain; mcev3@hotmail.com (M.C.E.-V.); mgraupera@csdm.cat (M.G.); oarias@csdm.cat (O.A.); aelvira@csdm.cat (A.E.); 2Gastrointestinal Physiology Laboratory. CIBERehd CSdM-UAB, Hospital de Mataró, 08404 Mataro, Spain; amartinma@csdm.cat (A.M.-M.); mbolivar@csdm.cat (M.B.-P.); oortega@csdm.cat (O.O.); 3Centro de Investigación Biomédica en Red, Enfermedades Hepato-Digestivas (CIBERehd) Instituto de Salud Carlos III, 28029 Madrid, Spain; 4Internal Medicine, Hospital de Mataró, Consorci Sanitari del Maresme, 08304 Mataro, Spain; mcabre@csdm.cat; 5ResearchUnit, Fundació Salut del Consorci Sanitari del Maresme, Hospital de Mataró, 08304 Mataro, Spain; epalomera@csdm.cat

**Keywords:** swallowing disorders, elderly, dementia, alzheimer disease, respiratory infections, mortality, follow-up

## Abstract

The prevalence of older patients with dementia and oropharyngeal dysphagia (OD) is rising and management is poor. Our aim was to assess the prevalence, risk factors, and long-term nutritional and respiratory complications during follow-up of OD in older demented patients. We designed a prospective longitudinal quasi-experimental study with 255 patients with dementia. OD was assessed with the Volume-Viscosity Swallowing Test and a geriatric evaluation was performed. OD patients received compensatory treatments based on fluid viscosity and texture modified foods and oral hygiene, and were followed up for 18 months after discharge. Mean age was 83.5 ± 8.0 years and Alzheimer’s disease was the main cause of dementia (52.9%). The prevalence of OD was 85.9%. Up to 82.7% patients with OD required fluid thickening and 93.6% texture modification, with poor compliance. OD patients were older (*p* < 0.007), had worse functionality (*p* < 0.0001), poorer nutritional status (*p* = 0.014), and higher severity of dementia (*p* < 0.001) than those without OD and showed higher rates of respiratory infections (*p* = 0.011) and mortality (*p* = 0.0002) after 18 months follow-up. These results show that OD is very prevalent among patients with dementia and is associated with impaired functionality, malnutrition, respiratory infections, and increased mortality. New nutritional strategies should be developed to increase the compliance and therapeutic effects for this growing population of dysphagic patients.

## 1. Introduction

Oropharyngeal dysphagia (OD) is a serious condition among patients with dementia with a prevalence of 32%–45% when clinically assessed and 84%–93% when instrumentally assessed [1]. Dementia is associated with impaired functionality and disability and has a high physical, psychological, social, and economic impact on patients and caregivers [2]. According to WHO, the number of persons with dementia is currently 47.5 million and by 2030 it is expected to increase to 75.6 million and by 2050 to 135.5 million [3]. Our group has shown that OD is a risk factor for mortality in older people and can cause serious complications such as dehydration, malnutrition, and respiratory infections, including aspiration pneumonia (AP) [4]; the latter being the most common cause of death among patients with dementia [5,6]. In addition, studies have found that OD is a significant predictor of poor clinical outcomes by increasing length of hospital stay and the economic burden of hospitalized patients with dementia [2]. Dementia impacts on the nutritional status of the patient, producing several complications, such as weight loss that can lead to anorexia, feeding apraxia, and dysphagia [7]. Despite being high, prevalence of OD in patients with dementia is variable according to the type and severity of dementia and the cortical and/or subcortical lesions suffered by these patients that affect the neural control of swallowing [8]. In patients with Alzheimer’s disease, the most common type of dementia, even in the initial phases, can impair the mastication process and the sensory aspects of swallowing, delaying oral transit time [9].We recently found older patients with OD and dementia presented a high prevalence of videofluoroscopic signs of impaired safety and efficacy of swallow and a severe impairment in airway protection mechanisms and that time to laryngeal vestibule closure (LVC) ≥ 340 ms predicted unsafe swallow in patients with dementia [10]. Masticatory and self-feeding difficulties and feeding dependency are also highly prevalent among these patients [11].

The screening of OD and its management in patients with dementia should be mandatory in psychogeriatric units (PU) of hospitals and healthcare centers, as OD is associated with negative clinical outcomes and impaired quality of life in these patients [6]. The Volume-Viscosity Swallowing Test (V-VST) is a quick and safe clinical assessment tool developed by our group [12] and can be used to evaluate older persons at risk of OD. A recent study using V-VST found that OD was very frequent in community-dwelling older persons with dementia and was associated with dependency and frailty [13]. In another recent study, using the same clinical tool, we found a minimal-massive intervention (MMI) in hospitalized older patients with OD (without severe dementia), improved nutritional status and functionality, and reduced hospital readmissions, respiratory infections, and mortality [14]. However, a systematic clinical assessment of OD and its proactive management in the population with dementia has not been carried out.

The aim of this study was to assess, at 18 months follow-up, the prevalence, risk factors, and complications of OD in patients with dementia from a PU of an intermediate care hospital. 

## 2. Materials and Methods 

### 2.1. Study Design

We designed a prospective, longitudinal, quasi-experimental study on patients with dementia consecutively admitted to the PU of an intermediate care hospital (Hospital de St. Jaume i Sta. Magdalena, Mataró, Spain) with 18 months follow-up to describe the prevalence of OD and the complications during the admission and follow-up period. On discharge, families and caregivers were given recommendations and advice on fluid adaptation according to the results of the V-VST, a clinical assessment tool that uses three volumes and viscosities in a progression of increased difficulty to assess signs of impaired safety and efficacy of swallow, texture-modified foods using one of two different textures (blended and easy mastication), nutritional supplements for malnourished patients or those at risk of malnutrition according to the Mini Nutritional Assessment short form (MNA-sf) [15], and oral hygiene recommendations including tooth brushing and the use of antiseptic mouthwashes. Patients were followed up at 3, 6, and 18 months by telephone calls and by reviewing the electronic medical history to assess the compliance with the fluid and diet recommendations, respiratory complications (lower respiratory tract infections (LTRI) and AP), rate of hospital readmissions, and mortality.

### 2.2. Study Population 

We studied 255 patients with dementia that were prospectively and consecutively recruited from the PU of Hospital de St. Jaume i Sta. Magdalena, Mataró, Catalonia, Spain between June 2013 and July 2016. Inclusion criteria were: patients over 18 years old and patients with dementia discharged from the PU who signed the informed consent. Exclusion criteria were: death during hospital admission, enteral nutrition, and refusal to participate. All participants and/or proxy decision makers were informed about the study and signed the informed consent. The study protocol was approved by the Ethics Committee of the Hospital de Mataró (CEIC 7/13) and was conducted according to the principles and rules laid down in the Declaration of Helsinki and its subsequent amendments and the EU rules for clinical trials on humans (EU Clinical Trial Regulation (EU-CTR, EU No 536/2014)).

### 2.3. Dysphagia Assessment

All participants were clinically assessed for OD with the V-VST during admission. The V-VST is a validated clinical assessment tool that uses three volumes (5, 10, and 20 mL) and viscosities (nectar, liquid, and pudding) together with a pulse-oximeter (to detect silent aspirations) to evaluate clinical signs of impaired efficacy and safety of swallow [12,16]. Diagnostic sensitivity and specificity for OD are 90% and 80%, respectively, and the reliability of V-VST is also high with an overall Kappa value of 0.77 (95% CI 0.65-0.89) [12].

### 2.4. Clinical Assessment of Patients with Dementia

We collected data on functional capacity with the Barthel Index that classifies the patients from 0 (totally dependent) to 100 (totally independent) points according to their capacity or inability to perform the activities of daily living (ADL) [17]; comorbidities with the Charlson Comorbidity Index that predicts the ten-year mortality for a for a patient who may have several comorbid conditions, the higher the score, the higher the mortality probability [18]; evaluation of dementia was performed by the study geriatrician and included: dementia type (Alzheimer’s disease, vascular dementia, etc.), severity with the Global Deterioration Scale (GDS) that goes from 1 (no cognitive alteration) to 7 (very severe cognitive alteration) [19], and the Functional Assessment Staging Test (FAST) that also goes from 1 (normal adult) to 7 (severe dementia) [20], and cognitive impairment measured with the Mini Mental State Examination (MMSE). We accordingly classified our patients into three levels (<11: severe; 11–20: moderate; and > 20: low/no cognitive deterioration) [21]. The MNA-sf was performed by a nutritionist and was used to assess nutritional status of the patients, classifying them into three categories: 0–7 points, malnourished; 8–12, at risk; 13–14, well-nourished [15]. Finally, we used the simplified Oral Hygiene Index–Simplified (OHI-S) to assess oral health status. A score lower than 1.1 points was considered good, between 1.1–3.0 fair, and between 3.1–6.0 poor oral health. The OHI-S is composed of the Debris Index (DI-S) that goes from 0 to 3 and measures the soft biofilm on the tooth surface and the Calculus Index (CI-S) (mineralized debris), from 0 to 3. The final score is the sum of both [22]. In addition, we registered number of functional teeth and edentulism.

### 2.5. Management of OD During Admission

During initial admission procedures (24–48h), trained nurses from the PU performed the V-VST and adapted: (1) fluids (volume and viscosity) to provide a safe and effective swallow; (2) solids (texture-modified food) according to swallowing and feeding evaluation by a dietitian using two different textures–blended diet and easy mastication; and (3) oral hygiene with an individualized protocol that included toothbrushing 3 times/day and antiseptic mouthwashes 2 times/day (chlorhexidine 0.12%) to reduce the oral bacterial load and the risk of respiratory infections (LTRI, AP). When patients had dentures, they were removed and placed in a disinfectant solution overnight; they also performed antiseptic mouthwashes twice a day (chlorhexidine 0.12%).

### 2.6. Adaptation of Fluid Viscosity

Fluid viscosity was adapted into three different levels according to the results obtained from the V-VST: thin liquids for patients without risk of aspiration (mineral water, milk, orange juice, etc.), in a range < 50 mPa·s, nectar viscosity and pudding viscosity. These lasts two levels were obtained using a thickener to increase their viscosity.

Nectar viscosity was obtained by adding 4.5 g of thickener (Resource Thicken Up, Barcelona, Spain) to 100mL mineral water to obtain a shear viscosity of 98.61 mPa·s ± 3.78 (100 mPa·s) at a shear rate of 50 s^-1^. Likewise, pudding viscosity was prepared by adding 9 g of thickener to 100 mL mineral water and achieved a viscosity of 4539.5 mPa·s ± 530.93 (4500 mPa·s). This specific thickening agent was affected by salivary amylase and both levels of viscosity were below 50 mPa·s (thin liquid) when analyzed after an oral incubation of 30 seconds.

### 2.7. Texture Modified Diets

Two different textures, blended and easy mastication, were used. Blended diet was defined as a smooth and uniform consistency (e.g., vegetable purée). Easy mastication diet consisted of soft dishes or moist food which can be broken into pieces easily (e.g., omelet). Modified texture diets were composed in 2 weekly menus: 2 for blended diet and 2 for easy mastication diet. Regarding the nutritional content, all the prescribed diets were normocaloric and normoproteic, corresponding to the following ranges: 1700–1800 kcal/day and 60–70 g of protein/day. Diets were supplemented, if needed. In order to increase the caloric content by 150 kcal per day, three high-caloric foods were given, such as oil and honey. In contrast, if the protein content needed to be increased, three high protein foods were selected to add between 16–18 g of protein per day such as fish or eggs. Nutritional supplements were prescribed according to accepted clinical practice at this institution [14].

### 2.8. Recommendations on Discharge

Families and caregivers were given personalized recommendations on discharge based on the treatment provided during admission. Recommendations consisted of four blocks of information: (1) adaptation of fluids (volume and viscosity); (2) texture adaptation (texture-modified foods) as previously described (easy mastication, blended diet and a combination of both diets); (3) nutritional supplementation with natural additives in patients at risk of malnutrition or malnourished (MNA-sf); and (4) oral hygiene recommendations according to the results of the OHI-S, including toothbrushing and the use of chlorhexidine mouthwashes). During admission, nurses carried out intense health education at the bedside for family members and caregivers on these main topics. 

### 2.9. Follow-Up Period

Follow-up visits were performed by telephone interviews with families and caregivers and review of the clinical outcomes and complications registered in the electronic medical history (primary care and hospital recordings) at 3, 6, and 18 months after discharge. Individual clinical history allows access to all the medical information of a patient and can be consulted from any health center in Catalonia under security and confidentiality protocols. Compliance with the recommendations was checked with a specially designed questionnaire. The outcomes associated with OD like functional status, LTRI, hospital readmissions and survival were registered. 

### 2.10. Statistical Analysis

Qualitative data were presented as relative and absolute frequencies and analyzed by the Fisher’s exact test or the Chi-square test. Continuous data were presented as mean ± standard deviation (SD) and compared with the T-test (intergroup comparisons) or Paired T-test (intragroup comparisons); for those variables that did not follow a normal distribution, we used the nonparametric Mann–Whitney U-test (intergroup comparisons) or the Wilcoxon-paired test (intragroup comparisons). The cumulative incidence of respiratory infections was calculated taking into account the total patients who suffered respiratory infections (LTRI and pneumonia) over 18 months/total number of patients. To analyze the adjusted effect of OD on complications, we used a Multivariate Cox Regression for mortality and Multivariate Logistic Regression for respiratory infections including LRTI and pneumonia. In these models, OD was adjusted for age, sex, functional capacity (Barthel index), nutritional status (MNA-sf), and cognitive deterioration (GDS). Results were interpreted according to the obtained *p* value, the magnitude of the observed effect, and their clinical and biological plausibility. Statistical significance was accepted if *p* values were less than 0.05.

## 3. Results

### 3.1. Hospitalization Period

#### 3.1.1. Demographic and Clinical Characteristics of Study Population

Two hundred and fifty-five older patients with dementia were recruited (61.6% women) with a mean age of 83.5 ± 8 years. The main cause of dementia in the study population was Alzheimer’s disease (52.9% (135)), followed by mixed dementia (25.1% (64)), Lewy’s bodies dementia (5.5% (14)), mild cognitive impairment (5.1% (13)), vascular dementia (3.1% (8)), Parkinson’s disease associated dementia (1.6% (4)), and other causes (6.7% (17)). Up to 60% (153) of study patients presented a GDS ≥ 6 (severe dementia). A Mini Mental State Examination (MMSE) showed that the majority of patients were in a moderate–severe cognitive impairment (87.4%). Mean Charlson score was 2.01 ± 1.4 points and Barthel Index was very low, 30.8 ± 24.7on admission and 39.6 ± 26.5 on hospital discharge. Up to 47.8% (122) of patients were admitted to the Psychogeriatric Unit of the hospital, 44.7% (114) from home and 7.5% (19) from nursing homes. 

#### 3.1.2. Oropharyngeal Dysphagia

Up to 85.9% (219) patients with dementia presented signs of OD according to the V-VST. Signs of impaired efficacy vs. safety of swallow were found in 83.1% (182) and 81.7% (179) of patients, respectively, and 64.8% (142) presented both impairments. Patients with OD were older (84.06 ± 7.8 vs. 80.16 ± 8.5; *p* = 0.007), with poorer functional capacity on admission (28.65 ± 24.36 vs. 44.03 ± 23.07; *p* < 0.0001) and greater severity of dementia (< 0.0001) than non-dysphagic subjects (ND) (Table 1). We found no differences between type of dementia and OD prevalence.

Results from the V-VST showed higher prevalence of unsafe swallows at liquid viscosity (60.8%) (*p* < 0.0001 vs. nectar and pudding). The safest viscosity was pudding (89.0%), presenting significant differences when compared with liquid for all volumes (*p* < 0.0001 for 5 and 20 mL and *p* < 0.01 for 10 mL). We also found significant differences when comparing nectar vs. liquid at 5 mL volume (*p* < 0.0001). No relationship was observed between volume and impaired safety signs (Figure 1). Prevalence of signs of impaired efficacy of swallow remained stable with the tested viscosities (nectar 65.3%; liquid 56.3%; and pudding 64.3%; ns). On the other hand, signs of impaired efficacy increased with volume, being more prevalent with 20 mL. We found statistically significant changes regarding residue when we compared nectar vs. liquid (10 and 20 mL; *p* < 0.05) and pudding vs. liquid (20 mL; *p* < 0.01) (Figure 1).

Following the V-VST results, only 17.3% (38) of patients could safely drink thin liquid (<50 mPa·s), 55.2% (121) needednectar (100 mPa·s) and 27.5% (60) pudding viscosity (4500 mPa·s) to achieve a safe swallow. Regarding texture modified diets, 53.4% (117) of patients needed easy mastication diet, 40.2% (88) blended diet, 5.9% (13) mixed diet (combining both diets), and only 0.5% (1) could take a normal diet without a texture adaptation. Up to 31% (79) of patients were totally feeding dependent. 

#### 3.1.3. Nutritional Status

Mean weight on admission was 62.6 ± 12.2 kg and the BMI was 26.7±1.2 kg/m^2^. According to MNA, 51.6% (127) of study patients were malnourished, 44.7% (110) were at risk of malnutrition, and only 3.7% (9) presented a normal nutritional status. Mean MNA-sf in patients with OD was significantly lower than ND (OD 7 ± 2.7 vs. ND 8.2 ± 2.5; *p* = 0.014), however, there were no differences in categorized nutritional status between both groups (Table 1). 

#### 3.1.4. Oral health and Hygiene Status

According to OHI-S, patients showed fair mean oral health score (2.7 ± 1.2), with a debris index of 1.4 ± 0.8 and calculus index of 1.3 ± 1.0. Regarding OHI-S categorization, 27.6% (24), 56.3% (49), and only 16.1% (14) of patients presented a poor, fair, and good OHI-S, respectively. Of the study participants, 40.1% (65) had less than 10 teeth and 31.4% (59) used dentures. On discharge, after nursing oral health intervention, the OHI-S significantly improved (1.7 ± 1.1; *p* < 0.0001) due to a significant debris score reduction (0.7 ± 0.7; *p* < 0.0001) as the calculus index did not improve. There were no main differences in oral health and hygiene status between OD and ND patients on admission and discharge.

#### 3.1.5. Clinical Outcomes during Hospitalization

The average stay in PU was 54.8 ± 68.4 days (53.3 ± 61.2OD vs. 63.7 ± 102.7 ND; ns). The most common pathologies found in the study population were arterial hypertension (66.7%), diabetes mellitus (27.5%), atrial fibrillation (18.4%), and stroke (12.2%). The most frequent reason for admission was the psychological and behavioral symptoms associated to dementia (57.5%). During admission, these symptoms were slightly more frequent in ND patients (88.1% OD vs 97.2% ND; *p* = 0.100) and those who had a higher non-significant prescription of psychoactive drugs (2.8 ± 1.6 OD vs. 3.3 ± 1.3 ND; *p* = 0.088). The incidence of LTRI during admission was similar in both groups (total LTRI events/total study population (0.2 ± 0.5 OD vs. 0.1 ± 0.3 ND; ns), although there was a trend toward a higher rate of other general complications in the OD group (total general complications events/total study population (0.6 ± 0.9 OD vs. 0.3 ± 0.5 ND; *p* = 0.068), urinary infection being the most common affect over the whole study population (16.5% (42)). On discharge, 54.1% (138) of patients went home (52.5% OD vs. 63.9% ND; ns) and 38% (97) to a nursing home (47.5% OD vs. 36.1% ND; ns). 

### 3.2. Follow-Up Period—18 months after Discharge

#### 3.2.1. Treatment Compliance

During the follow-up period, up to 11.7% of OD patients did not follow the recommendations about texture diet adaptation and up to 40.6% did not follow fluid adaptation recommendations due to the caregiver’s difficulty to adapt textures and viscosities (17.9% and 38.0%, respectively). The most frequent reason for not following recommendations was the patient’s rejection of high viscosity and/or textures. The main factors that influenced compliance during follow-up were the presence of non-compliance during admission (*p* < 0.001) and reported difficulties in preparing the recommended diet textures (*p* = 0.055) and high viscosity of fluids (*p* < 0.0001). 

#### 3.2.2. Long Term Clinical Outcomes 

The main complications of patients with OD during follow-up were LTRI and pneumonia that was diagnosed in 43.0% (99) and 12.6% (28) of patients, respectively. Up to 21.2% (49) were readmitted with respiratory infections. Finally, 72 patients died (10.5% (23) at 3 months, 8.2% (16) at 6 months, and 18.3% (33) at 18 months). 

When we compared the long-term clinical outcomes of patients with and without OD after 18 months of follow-up, the cumulative incidence of respiratory infection (LTRI and pneumonia) during the study period was higher in OD patients (total respiratory infection events during follow-up/total study population (OD 1.0 ± 1.5 vs. ND 0.6 ± 1.2; *p* = 0.040)) as well as its prevalence (total patients who suffered respiratory infections/total study population (OD 51.8% vs. ND 27.8%; *p* = 0.011)). We observed significant differences between both groups for LTRI (OD 45.9% vs. ND 27.8%; *p* = 0.046) but not for pneumonia (OD 14.0% vs. 5.6%; *p* = 0.270). In this period, episodes of hospital readmissions were more frequent in ND patients (*p* = 0.015), however, this difference was not significant for respiratory infections (*p* = 0.828) or for readmissions for other diseases (*p* = 0.075) (Table 2). During follow-up, OD patients had significantly higher mortality than ND (32.9% OD vs. 13.9% ND; *p* = 0.0002) (Figure 2). After multivariate analysis adjusted for confounding factors (age, sex, functional status, and dementia severity), we observed that having OD was closely associated with mortality (OR-2.8 (95%CI 0.972–8.106; *p* = 0.056) as well as respiratory infections (OR-2.4 (95%CI 0.958–5.79; *p* = 0.062). In addition, age was independently associated with mortality and respiratory infection during the follow-up period (OR-1.04 (95%CI 1.012–1.075); *p* = 0.012 and OR-1.05 (95%CI 1.004–1.088); *p* = 0.029) and functionality was independently associated with mortality (OR-0.98 (95%CI 1.012–1.075); *p* = 0.012) (Table 3).

## 4. Discussion

This study clearly shows that patients with dementia admitted to the PU of an intermediate care hospital had a very high prevalence of OD (85.9%), malnutrition (MN) (51.6%), and poor or fair oral health status (83.9%). Risk factors for OD were older age, functional dependence, and severity of dementia according to GDS/FAST. In these patients, OD was associated with a greater risk of being malnourished than those without OD on admission, and with respiratory infections and mortality during the 18-month follow-up. Patients with OD and dementia required high viscosity fluids and texture-modified diets to provide a safe swallow but compliance with these treatments was very poor as well as their clinical outcomes. These results might suggest that the therapeutic effect of this strategy is weak and therefore the nutritional management of OD patients with dementia must be improved, but they also suggest that OD management in severely demented patients must be close to palliative care as we cannot change the severity and natural history of OD in these patients.

We included very old OD patients with dementia in this study. Previous studies have also reported similar age ranges as the age of these institutionalized patients with dementia is generally very advanced [6,23,24]. Regarding comorbidities, the Charlson Index score in our population was similar to that reported by other authors from a PU and in a group of outpatients with dementia [23,25]. Likewise, similar impairments in functional capacity were found in a previous study [25]. We also found that OD was associated with impaired functional status, indicating that functionality is a key factor in the pathophysiology of OD, as described previously [26,27]. Alzheimer’s was the most frequent type of dementia, also found in previous studies [23,25]. In addition, those patients with OD were in a significantly more severe phase of dementia. One publication reported that, as dementia progresses with worsening cognitive and functional status, eating behavior deteriorates, affecting up to 77% of institutionalized patients with a GDS-FAST > 6 [7]. This data highlights the progressive swallowing difficulties suffered by patients with dementia, indicating that OD is a major clinical concern in this population at these advanced stages of age and evolution of this disease. 

There is a higher incidence of aspiration in patients with more severe dementia [28]. Our prevalence of OD is higher than that found by other authors that used instrumental assessments [29,30] and similar to that in one study of older patients with dementia from an outpatient clinic assessed also with the V-VST (85.9% vs. 86.6%) [13]. However, in the latter study, only 8.9% of OD patients presented severe dementia (68.2% in ours) and fewer signs of impaired safety of swallow (46.40% vs. 81.7% in ours). Despite similar OD prevalence, the differences found in safety impairment between both studies may be due to the greater severity of dementia in our population. OD is also a major geriatric syndrome because it is highly prevalent in old age, multifactorial, associated with multiple comorbidities and bad prognosis, and is only treatable when a multidimensional approach is applied, including the participation of a multidisciplinary team [31]. Aspiration pneumonia (AP) and respiratory infections are common, severe complications of OD in these patients. The pathophysiology of AP includes three main risk factors: (1) OD with impaired safety of swallow (aspirations); (2) frailty and impaired health status; and (3) poor oral health and hygiene with colonization by respiratory pathogens [32]. Accordingly, we developed the minimal-massive intervention (MMI) that aims to treat these three risk factors and consists of: (a) fluid thickening according to the V-VST to avoid impaired safety of swallow and aspirations; (b) texture modification and nutritional supplementation (caloric/proteic) according to MNA-sf to improve nutritional and health status; and (c) oral hygiene (tooth brushing and mouthwashes) to reduce colonization by respiratory pathogens. In an MMI pilot study on a group of OD older patients in a less severe clinical condition, we found a reduction in hospital readmissions, LTRI incidence, and mortality (treated group vs. control) [14]. Compared to our patients, those in the MMI study were similar in age but they presented less severe impairment in functionality (Barthel 28.7 ± 24.4 vs. MMI 59.5 ± 26.8; *p* < 0.0001). In addition, it should be taken into account the advanced stage of dementia of the patients in the present study and their severe cognitive deterioration. All this data shows the high level of impairment of our study patients which could explain the severe complications and poor prognosis they had despite following a similar management strategy for OD. Taken all together, we believe that this specific kind of phenotype of advanced demented patients probably needs a palliative treatment rather than a compensatory one. The phenotype of patients and the natural history of their condition is the main variable that should drive the selection of active, compensatory, or palliative treatments for their swallow dysfunction.

As we have described previously, in this study, we observed an increase in safe swallow when viscosity increased. Impairments in safety of swallow increased only with the highest volume assessed except for pudding viscosity. In contrast, impaired efficacy was more prevalent with increased volume [33,34]. The most common recommendation on discharge from the PU (55.2%) was to adapt fluids to nectar viscosity (100m Pa·s). However, prescribed viscosity can decrease sharply due to the effect of oral salivary amylase (< 50mPa·s) in this phenotype of patients with high prevalence of impairments in the oral phase (prolonged). This enzyme breaks the O-glycoside bonds of starch which causes a loss of the therapeutic effect, suggesting we can improve our results by using an amylase-resistant gum-based thickener [35]. Regarding nutrition, only 3.66% of study patients presented a proper nutritional status and the rest were at risk of MN or malnourished. Altogether, these results raise the question of whether malnutrition can be reversed in older patients with dementia and OD and suggest that a decision between palliative vs. compensatory nutritional management should be taken. High rates of MN have been previously described in hospitalized older patients and very old patients with OD (45.3% and 36.8%) and we also found that OD is a risk factor for MN with an OR of 1.6 [27,36]. In our study, the most common recommendation on discharge was to provide the easy mastication/fork mashable diet (53.4%). However, non-compliance with this recommendation was 11.7%, although the least compliance was with fluid adaptation. Adherence to treatment (fluids and diet) in our study population was only 52.4% during the entire follow-up period and it is known that compliance with chronic treatments ranges from 40%-50% and short-term treatments from 70%–80% [37,38]. Our study was conducted with a modified starch thickener; however, new gum thickeners have better rheological properties (amylase resistance and stability) [34,39,40]. OD patients, especially those who required pureed texture diet, had a higher risk of MN due to the appearance of the food and its unpleasant organoleptic characteristics (color, smell, presentation) which can lead to a loss of pleasure while eating, further aggravating their poor nutritional status. Diets used for the study were developed according to the British Dietetic Association guidelines on Dysphagia Diet Food Texture Descriptors [41]: blended diet was divided into two textures, B and C (e.g., soft whipped cream and mousse, respectively); easy mastication diet also became two textures: D and E texture (e.g., flaked fish in thick sauce and tender meat casseroles). Nonetheless, adapting texture and nutritional content was not enough to improve adherence, and thus, we recently developed the Triple Adaptation of Diet [42]. These new diets consist of: (a) organoleptic, (b) calorico-proteic, and (c) rheological (texture) adaptation of Mediterranean dishes designed to increase compliance and reduce MN and dehydration in OD patients. Recipes were developed to meet patients’ needs and to be reproducible at patients’ homes, which may improve their clinical outcomes. A pilot study in our hospital suggested a strong therapeutic effect of these diets in OD older patients [14], though we also suggested that there was a need for better quality evidence regarding this promising intervention [43]. Recently, Hospital of Mataró and Fundation Furega developed a full set of menus based on the Mediterranean diet, with 2 textures and 2 caloric/proteic adaptations to increase compliance and the pleasure of eating [42]. 

The third point of our intervention was oral health and hygiene (OH). Patients with dementia, especially in advanced phases, often have poor oral health and OH and missing teeth [44,45]. Ortega et al. described the oral health of older patients with and without OD according to the OHI-S and found that both groups had poor OH but that it was significantly worst in OD patients and a higher risk of AP [46]. Our patients showed poor OH but there were no differences between groups, maybe due to their advanced stages of dementia and functional dependence. The implementation of new protocols of oral health in the PU improved OH and were translated into significant differences when comparing the OHI-S on admission vs. discharge. These improvements show that with basic tooth brushing and antiseptic mouthwashes use, the accumulation of bacterial biofilm on the dental surfaces is reduced, and consequently, the risk of respiratory infections is reduced, as OH has been previously related to the colonization of respiratory pathogens [47,48]. Thus, OH should form part of the basic management algorithm of OD patients with dementia and strategies to monitor compliance must be developed.

Finally, the main complication during the study in our OD patients was respiratory infections, with significantly higher cumulative incidence at 18 months compared with ND patients (OD 0.99 ± 1.48 vs. ND 0.6 ± 1.2; *p* = 0.040). Similar results for LTRI, in this case by annual incidence rate, were described among older OD patients living in the community (OD 40.0% vs. ND 21.8%; *p* = 0.030) [49]. However, we found more hospital readmission in ND than in the OD group. A possible explanation for this is that more OD patients were more institutionalized on hospital discharge due to their more severe dementia. Most of their complications could be managed by medical staff in nursing homes and long-term centers without the need to be transferred to the acute care hospital, also in terms of therapeutic adequacy. One of the most significant results of the study was that mortality over the 18-month follow-up was significantly higher in OD than in ND patients (32.9% vs. 13.8%; *p* = 0.002) and, after multivariate analysis, OD was also closely associated with mortality (OR-2.8) and respiratory infections (OR-2.4). In PU patients, studies have shown that mortality is associated with dysphagia [26,27,36], functional dependence [6], malnutrition [36], and severity of dementia [6]. We have found, however, that non-compliance with recommendations is not associated with higher mortality or greater number of complications, again questioning the therapeutic efficacy of our compensatory interventions in this extremely ill population. Perhaps the effect of treatment adherence on mortality is not appreciable because the sample is small and, probably, the high severity of dementia and functional dependence have more weight on this outcome than the potential therapeutic effect of our treatment. A study found that the most common complications in patients with very advanced dementia were pneumonia and feeding problems and that aggressive treatment did not influence the prognosis of patients. In addition, complications were associated with high mortality rates in the following months [6]. Perhaps for that reason, and based on the results found, we could consider older OD patients with advanced dementia as palliative patients with regard to the therapeutic effort and the need for diagnostic tests (clinical vs. instrumental assessment). Attending the poor prognosis of this population, their needs, and dementia severity, a simplified MMI with less monitoring but including the adaptation of fluids (gum-based thickeners) and diet (triple adaptation of solids), and an adequate basic maintenance of oral health is proposed. The main aim of this MMI is to reduce as much as possible the risk of developing OD-associated complications, [50] and thus, improve patients’ and carers’ quality of life. The main limitation of the study was that we did not include separate scales to show clinical, behavioral, and functional status to comprehensively describe dementia.

## 5. Conclusions

OD is a prevalent condition in patients with advanced dementia and is associated with severe complications, poor prognosis, and low compliance with fluid and texture adaptation. Taking into account these findings, we propose a management plan for these patients according to the severity of dementia. For the early–moderate stages, we suggest using a multimodal treatment with a periodic on-site thorough monitoring based on the MMI with better thickening agents, texture modified diets, natural supplements, and nutritional products with high palatability to improve adherence and patients’ prognosis. In the more severe stages of dementia, with worst functionality and high rates of complications including mortality, a palliative MMI management with a less exhaustive follow-up is proposed to reduce the rate of complications related to OD. Supported by the study results, we consider that new nutritional strategies should be developed to increase patient compliance and its therapeutic effects in the clinical management of OD patients with dementia and that new evidence regarding these promising interventions should be provided.

## Figures and Tables

**Figure 1 nutrients-12-00863-f001:**
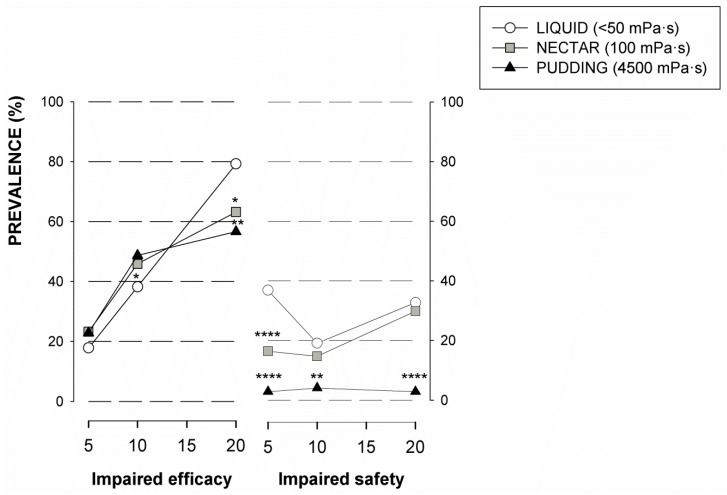
Prevalence of patients with impaired signs of efficacy and safety according to the different levels of viscosity included in the Volume-Viscosity Swallowing Test (V-VST). * *p* < 0.05; ** *p* < 0.01; *** *p* < 0.001; **** *p* < 0.0001 vs. liquid.

**Figure 2 nutrients-12-00863-f002:**
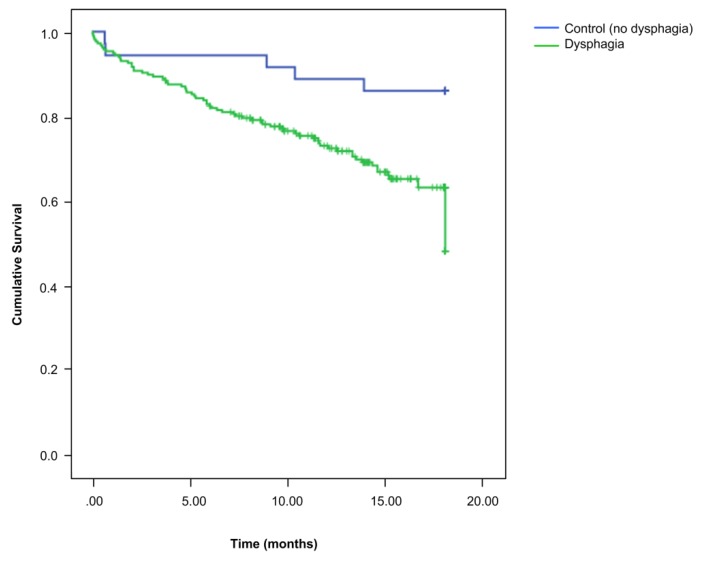
Eighteen-month survival curves for dysphagic and non-dysphagic patients. OD: oropharyngeal dysphagia patients; ND: patients without OD.

**Table 1 nutrients-12-00863-t001:** Demographic characteristics of patients with and without dementia included in the study.

	TOTAL (*n* = 255)	OD (*n* = 219)	ND (*n* = 36)	*p*-Value
Age (years)	83.5 ± 8	84.06 ± 7.8	80.16 ± 8.5	**0.007**
Sex (♀)	61.6% (157)	61.2% (134)	63.9% (23)	0.757
Charlson	2.01 ± 1.4	2.1 ± 1.4	1.8 ± 1.1	0.432
Barthel Admission Discharge	30.8 ± 24.7 39.6 ± 26.5	28.65 ± 24.4 37.46 ± 26.6	44.03 ± 23.1 52.64 ± 21.7	**< 0.0001** **0.002**
Dementia severity GDS (% (*n*)) 2–3 4 5 6 7 FAST (% (*n*)) 2–3 4 5 6 7 Cognitive assessment MMSE MMSE (% (*n*)) < 11 11–19 > 19	*n* = 233 1.6 (4) 9.0 (23) 20.8 (53) 42.7 (109) 17.3 (44) *n* = 228 0.4 (1) 8.3 (19) 14.9 (34) 50.4 (115) 25.9 (59) *n* = 191 11.9 ± 7.0 38.2 (73) 49.2 (94) 12.6 (24)	*n* =2 01 0.5 (1) 9.5 (19) 21.9 (44) 46.3 (93) 21.9 (44) *n* = 200 0 (0) 8 (16) 15.5 (31) 48.0 (96) 28.5 (57) *n* = 161 11.7 ± 7.1 38.5 (62) 49.7 (80) 11.8 (19)	*n* = 32 9.4 (3) 12.5 (4) 28.1 (9) 50.0 (16) 0.0 (0) *n* = 28 3.6 (1) 10.7 (3) 10.7 (3) 67.9 (19) 7.1 (2) *n* = 30 12.1 ± 6.8 36.7 (11) 46.7 (14) 16.7 (5)	**< 0.001** **0.007** 0.759 0.761
MNA-sf	7.17 ± 2.67	7 ± 2.68	8.2 ± 2.45	**0.014**
MNA-sf (% (*n*)) Well-nourished (12–14) At risk (8–11) Malnourished (0–7)	*n* = 255 3.7 (9) 44.7 (110) 51.6 (127)	*n* = 211 3.3 (7) 43.1 (91) 53.6 (113)	*n* = 35 5.7 (2) 54.3 (19) 40.0 (14)	0.305
OHI-S DI-S CI-S	2.7 ± 1.2 1.4 ± 0.8 1.3 ± 1.0	2.7 ± 1.2 1.4 ± 0.9 1.4 ± 1.0	2.5 ± 1.1 1.4 ± 0.5 1.1 ± 0.7	0.540 0.790 0.410

OD: oropharyngeal dysphagia patients; ND: patients without OD; *n*: number of patients; GDS: Global Deterioration Scale; FAST: Functional Assessment Staging Test; MMSE: Mini Mental State Examination; MNA-sf: Mini Nutritional Assessment short form; OHI-S: Oral Health Index-Simplified; DI-S: Debris Index simplified; CI-S: Calculus Index Simplified. Note: bold values are statistically significant *p* < 0.05.

**Table 2 nutrients-12-00863-t002:** Clinical outcomes at 18 months follow-up on patients with and without dysphagia.

	18 MONTHS FOLLOW-UP			
	TOTAL (*n* = 255)	OD (*n* = 219)	ND (*n* = 36)	*p*-Value
Resp. Infections (episodes/patient) Resp. Infections (% (*n*)) LTRI (% (*n*)) Pneumonia (% (*n*)) Fever without focus (% (*n*))	0.9 ± 1.5 48.1 (112) 43.0 (99) 12.6 (28) 6.4 (14)	1.0 ± 1.5 51.8 (102) 45.9 (89) 14.0 (26) 7.7 (14)	0.6 ± 1.2 27.8 (10) 27.8 (10) 5.6 (2) 0.0 (0)	**0.040** **0.011** **0.046** 0.270 0.134
Re-admissions (episodes/patient) Re-admissions (% (*n*)) Respiratory infections (% (*n*)) Other diseases (% (*n*))	0.6 ± 1.0 44.3 (104) 21.2 (49) 30.9 (69)	0.6 ± 0.9 41.7 (83) 21.0 (41) 28.3 (53)	0.9 ± 1.0 58.3 (21) 22.2 (8) 44.4 (16)	**0.015** 0.071 0.828 0.075
Institutionalization rate (% (*n*))	45.9 (117)	47.5 (104)	36.1 (13)	0.118

OD: oropharyngeal dysphagia patients; ND: patients without OD; *n*: number of patients; LTRI: lower track respiratory infection. Note: bold values are statically significant *p* < 0.05.

**Table 3 nutrients-12-00863-t003:** Adjusted effect of oropharyngeal dysphagia on mortality and respiratory infections respectively (multivariate analysis).

	OR (95% CI)	*p*-Value
Mortality during follow-up		
Oropharyngeal dysphagia	2.81 (0.972–8.106)	0.056
Age	1.04 (1.012–1.075)	0.012
Barthel score	0.98 (0.971–0.994)	0.004
Dementia severity	1.09 (0.552–2.160)	0.801
Respiratory Infection during follow-up		
Oropharyngeal dysphagia	2.36 (0.958–5.793)	0.062
Age	1.05 (1.004–1.088)	0.029
Barthel score	1.00 (0.984–1.013)	0.823
Dementia severity	0.93 (0.438–1.965)	0.846

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
