# Peer review of "Prevalence, Risk Factors, and Complications of Oropharyngeal Dysphagia in Older Patients with Dementia"

_nutrients, 2020, doi:10.3390/nu12030863_

Round 1

Reviewer 1 Report

Prevalence, risk factors and complications of oropharyngeal dysphagia in older patients with dementia

This manuscript describes a prospective longitudinal quasi-experimental study in older adults with dementia which aims to assess the prevalence, risk factors and long term nutritional and respiratory complications during follow up of OD over 18 months.

This is a nicely presented manuscript. Some minor comments are provided below for consideration by the authors, especially with regards to the discussion section.

Introduction

  • L 64 could you please indicate briefly here what the V VST involves to assist the author here.

Methods

  • Could some information be provided for the Charlson Comorbidity Index, Barthel, FAST and GDS on how to interpret the results. This is presented well for the MNA and OHIS but not the tools on L107-8.
  •  

Results

  • Nicely presented
  • Consider only reporting to one decimal place throughout – this is done inconsistently (sometimes 1 sometimes 2 decimal places).
  • L 222 – typo extra S

Discussion

  • L 290 – typo MN should be MNA, consider writing malnutrition and risk of malnutrition (otherwise introduce MN as a new acronym).
  • L 292 – MN consider writing great risk of being malnourished than those without OD
  • L322 – typo foundin
  • L326 – could you describe what an optimal multidimensional approach would entail?
  • L357 typo
  • L363-4 consider re-writing this sentence, it is not currently clear
  • L 381 – please introduce what BDI guidelines

General comments

  • Consider referring to this group as patients with dementia consistently throughout, sometimes it is dementia patients

Author Response

Please, see also the response attached in PDF.
This manuscript describes a prospective longitudinal quasi-experimental study in older adults with dementia which aims to assess the prevalence, risk factors and long term nutritional and respiratory complications during follow up of OD over 18 months. This is a nicely presented manuscript. Some minor comments are provided below for consideration by the authors, especially with regards to the discussion section.
Thank you for your revision and your positive comments to improve our paper.
Introduction
- L 64 could you please indicate briefly here what the V VST involves to assist the author here.
- To give more information about the V-VST we have added: "a clinical assessment tool that uses three volumes and viscosities in a progression of increased difficulty to assess signs of impaired safety and efficacy of swallow"..

Methods
- Could some information be provided for the Charlson Comorbidity Index, Barthel, FAST and GDS on how to interpret the results. This is presented well for the MNA and OHIS but not the tools on L107-8.
- We have added more information on the tests commented in order to facilitate the interpretation of the results. This is now described in lines 111 – 119: "We collected data on functional capacity with the Barthel Index that classifies the patients from 0 (totally dependent) to 100 (totally independent) points according to their capacity or inability to perform the activities of daily living (ADL); comorbidities with the Charlson Comorbidity Index that predicts the ten-year mortality for a patient who may have several comorbid conditions, the higher the score, the higher the mortality probability; evaluation of dementia was performed by the study geriatrician and included: dementia type (Alzheimer’s disease, vascular dementia, etc.), severity with the Global Deterioration Scale (GDS) that goes from 1 (no cognitive alteration) to 7 (very severe cognitive alteration), the Functional Assessment Staging Test (FAST) that also goes from 1 (normal adult) to 7 (severe dementia), and cognitive impairment measured with the Mini Mental State Examination (MMSE). We accordingly classified our patients into three levels (<11:severe; 11-20: moderate; and >20 low/no cognitive deterioration".  Thank you.

Results
- Nicely presented
- Thank you.
- Consider only reporting to one decimal place throughout – this is done inconsistently (sometimes 1 sometimes 2 decimal places).
- Presentation of data regarding decimals has been unified in all the document to1 decimal.
- L 222 – typo extra S
- Typographic mistake has been corrected.

Discussion
- L 290 – typo MN should be MNA, consider writing malnutrition and risk of malnutrition (otherwise introduce MN as a new acronym).
- We have introduced MN as a new acronym and your suggestion has been taken in to account, thank you.
- L 292 – MN consider writing great risk of being malnourished than those without OD
- We have written"great risk of being malnourished than those without OD" as suggested by the reviewer.
- L322 – typo foundin
- Typographic mistake has been solved.
- L326 – could you describe what an optimal multidimensional approach would entail?
- A multidimensional approach for the treatment of dysphagia should include the participation of a multidisciplinary team as dysphagia is a complex condition and many healthcare professionals can be involved in its management and treatment such as speech and language therapists, gastroenterologists, ENTs, surgeons, radiologists, nurses, dietitians, etc. To clarify this, we have added "including the participation of a multidisciplinary team" to the sentence.
- L357 typo
- Typographic mistake has been solved.
- L363-4 consider re-writing this sentence, it is not currently clear
- The sentence has been re-written as follows: "Altogether, these results raise the question whether malnutrition can be reversed in older patients with dementia and OD and suggest that a decision between palliative vs. compensatory nutritional management should be taken."
- L 381 – please introduce what BDI guidelines
- We have added: British Dietetic Association guidelines on Dysphagia Diet Food Texture Descriptors.

General comments
- Consider referring to this group as patients with dementia consistently throughout, sometimes it is dementia patients
- We have unified the criteria and referred to them as patients with dementia in the whole document. Thanks.

Reviewer 2 Report

Background:

The present manuscript aims at evaluating the prevalence, risk factors and long term complications of  oropharyngeal dysphagia (OD)  in 255 older demented patients with a mean age of 83.5 years.

OD was assessed with the Volume-Viscosity Swallowing Test and a comprehensive geriatric evaluation was performed at baseline.

The V-VST, developed and validated by the same Group, is an effort bedside test that uses boluses of different volumes and viscosities to identify clinical signs of impaired efficacy (impaired labial seal, piecemeal deglutition, and residue) and impaired safety of   (cough, voice changes, and oxygen desaturation ≥3%) and it has been previously validated against videofluoroscopy (VFS) as the reference standard, albeit in a different  type of patient sample.

In the study sample, prevalence of OD 25 was 85.9%. and it was associated with impaired functionality, malnutrition, respiratory infections and increased mortality after 18 months follow-up.

Comments:

OD is still a relatively neglected aspect of dementia management and here it was evaluated with a clinical tool, the V-VST. The V-VST promises to be a potentially useful instrument in this specific population because it has good psychometric properties with an easy-to-perform protocol which was designed to protect safety of patients, and has valid endpoints to evaluate safety and efficacy of swallowing and to detect silent aspirations.

By contrast, some methodological limitations refers to dementia assessment and are as follows:

- it is not clear who made and which criteria and tools were adopted for both dementia diagnosis and dementia type categorization.

- the Authors state that the main cause of dementia in their sample was Alzheimer’s disease which was recognized in 52.9% (135) of cases; however, they do not specify which additional subtypes of dementia were present in the reamining 48% of cases and how they were distributed.

- the assessment of dementia severity was made with the Global Deterioration scale (GDS) and the Functional Assessment Staging Test (FAST), but both scales – albeit largely clinically used - have raised doubts as sound research tools mostly for the qualitative heterogeneity of their component parts.

In particualr, the stages of the  Global Deterioration Scale are based on an  implicit theoretical assumptions about the linearity, temporality, and interdependence of cognitive, functional, and behavioral impairment in Alzheimer's disease.

By contrast, empirical evidence suggests that the psychiatric symptoms and the functional impairment occur earlier than predicted by the GDS, and that the rate of change is also different from that specified in the scale.

Separate scales to describe cognitive (i.e. Mini Mental State Examination or Montreal Cognitive Assessment), clinical (i.e. CDR sum of boxes), behavioral (i.e. Neuropsychiatric Inventory) and functional status (i.e. ADL and IADL scales) may be the best way to comprehensively describe dementia.

- it is not clear if OD prevalence and outcome associations are equal or different based on dementia type.

- Discussion appears too long and sometimes goes far beyond empirical data reported in the Results section: for example, it details  many of the interventions made on the basis of OD assessment, but these interventions were not an object of the study.

Author Response

Please, find also attached the pdf with the responses.
Comments and Suggestions for Authors
Background:
- The present manuscript aims at evaluating the prevalence, risk factors and long term complications of  oropharyngeal dysphagia (OD)  in 255 older demented patients with a mean age of 83.5 years. OD was assessed with the Volume-Viscosity Swallowing Test and a comprehensive geriatric evaluation was performed at baseline. The V-VST, developed and validated by the same Group, is an effort bedside test that uses boluses of different volumes and viscosities to identify clinical signs of impaired efficacy (impaired labial seal, piecemeal deglutition, and residue) and impaired safety of  (cough, voice changes, and oxygen desaturation ≥3%) and it has been previously validated against videofluoroscopy (VFS) as the reference standard, albeit in a different  type of patient sample. In the study sample, prevalence of OD 25 was 85.9%. and it was associated with impaired functionality, malnutrition, respiratory infections and increased mortality after 18 months follow-up.
Comments:
- OD is still a relatively neglected aspect of dementia management and here it was evaluated with a clinical tool, the V-VST. The V-VST promises to be a potentially useful instrument in this specific population because it has good psychometric properties with an easy-to-perform protocol which was designed to protect safety of patients, and has valid endpoints to evaluate safety and efficacy of swallowing and to detect silent aspirations.
- Thank you for your positive comments on the clinical assessment method we have developed and validated for OD.
By contrast, some methodological limitations refers to dementia assessment and are as follows:
- it is not clear who made and which criteria and tools were adopted for both dementia diagnosis and dementia type categorization.
- Dementia diagnosis was performed on all patients by the head of the Psychogeriatrics Unit, Dr. Carmen Espinosa, first author of the paper. In the manuscript we have added to the methods section (clinical assessment of patients with dementia): "evaluation of dementia was performed by the study geriatrician and included...".
- the Authors state that the main cause of dementia in their sample was Alzheimer’s disease which was recognized in 52.9% (135) of cases; however, they do not specify which additional subtypes of dementia were present in the reamining 48% of cases and how they were distributed.
- Thank you for this relevant question. Following your advice, we have included the different types of dementia of our study population. Accordingly, we have modified the demographical and clinical characteristics of study population: "The main cause of dementia in the study population was Alzheimer disease (52.9% (135)) followed by mixed dementia (25.1% (64)), Lewy's bodies dementia (5.5% (14)), mild cognitive impairment (5.1% (13)), vascular dementia (3.1% (8)), Parkinson's disease associated dementia (1.6% (4)) and other causes (6.7% (17))".
- the assessment of dementia severity was made with the Global Deterioration scale (GDS) and the Functional Assessment Staging Test (FAST), but both scales – albeit largely clinically used - have raised doubts as sound research tools mostly for the qualitative heterogeneity of their component parts. In particualr, the stages of the Global Deterioration Scale are based on an implicit theoretical assumptions about the linearity, temporality, and interdependence of cognitive, functional, and behavioral impairment in Alzheimer's disease. By contrast, empirical evidence suggests that the psychiatric symptoms and the functional impairment occur earlier than predicted by the GDS, and that the rate of change is also different from that specified in the scale. Separate scales to describe cognitive (i.e. Mini Mental State Examination or Montreal Cognitive Assessment), clinical (i.e. CDR sum of boxes), behavioral (i.e. Neuropsychiatric Inventory) and functional status (i.e. ADL and IADL scales) may be the best way to comprehensively describe dementia.
- We agree that the assessment of dementia could be improved and so we have added the MMSE cognitive deterioration scale that was collected in our study but not initially included in the manuscript. We have added it in the methods section, and the results (Table 1) to improve the description of dementia in our study patients and describe their cognitive status. We will take into account your suggestion for future studies when we need to evaluate dementia in order to have a comprehensive evaluation of the pathology. We have also added your comment as a limitation at the end of the discussion section: "The main limitation of the study was that we did not include separate scales to show clinical, behavioral and functional status to comprehensively describe dementia".
- it is not clear if OD prevalence and outcome associations are equal or different based on dementia type.
- Thank you for this comment but your question on whether the association of a specific prevalence of OD and clinical outcome was dependent on the type of dementia is beyond the scope of this manuscript.We have added the different types of dementia in the results section of the manuscript, but we found no differences between type of dementia and OD prevalence which why these data was not included in the manuscript (see the table attached in the pdf). We have added a comment in the results section (oropharyngeal dysphagia): We found no differences between type of dementia and prevalence of OD.
- Discussion appears too long and sometimes goes far beyond empirical data reported in the results section: for example, it details many of the interventions made on the basis of OD assessment, but these interventions were not an object of the study.
- Thank you, following your comments the discussion has been reduced by 20%. On the other hand, we believe that the description of these kinds of interventions, such as the minimal-massive intervention which was used in the current study and includes adaptation of fluids, texture-modified foods and oral hygiene, are key points for the management of these patients, and thus, it is essential to comment on them in the discussion section.
